# Risk Profile of Keratoconus among Secondary School Students in the West Region of Cameroon

Enowntai Nkongho Ayukotang [1,2,*], Vanessa Raquel Moodley [3] and Khathutshelo Percy Mashige [2]

1  Department of Biomedical Sciences, Faculty of Health Sciences, University of Buea, Buea P.O. Box 63, Cameroon
2  African Vision Research Institute, School of Health Sciences, University of KwaZulu-Natal, Private Bag X54001, Durban 4000, South Africa
3  Discipline of Optometry, School of Health Sciences, University of KwaZulu-Natal, Private Bag X54001, Durban 4000, South Africa
*  Correspondence: ayukotang@yahoo.com; Tel.: +237-674652651

**Abstract:** This study determined the risk factors of keratoconus (KC) among secondary school students in the West Region of Cameroon. A stratified, random sampling technique was used to select the 3015 secondary school students, 8 years and older, within the West Region of Cameroon. Selected school students completed the validated Keratoconus Risk Investigative Survey (KRIS) and a structured demographic questionnaire to determine the risk profile of KC. Descriptive analysis, logistic regression and *p*-values were used to provide an overview of the demographic findings and the risk factors of KC. Estimates were made as the proportion of affected school students and presented with a 95% confidence interval (CI). Multivariate logistic regression analysis was performed to explore the association between KC and the independent predictors that were found significant in the univariate analysis. The ages of the majority (93.2%) of students ranged from eight years to 18 years (mean = 13.18 ± years) and were mostly female (59.7%). Gender (OR 2.024, *p* < 0.001), eye rubbing (OR 3.615, *p* < 0.001), exposure to sunlight (OR 2.735, *p* < 0.001), blood relations with KC (OR 41.819, *p* < 0.001) and allergic experience (OR 1.070, *p* < 0.001) were considered. Eye rubbing was the most significant risk factor of keratoconus followed by refractive error, allergic experiences and sunlight exposure. These findings support the evidence that the etiology of KC is multifactorial, with eye rubbing being the most significant factor in this cohort. There is a need to address eye rubbing among students to minimize the risk of KC. Furthermore, 34.46% of students in Cameroon were at risk of developing KC. Hence the risk profile is that one engages in eye rubbing, has a family member with KC, spends more than eight hours per week in the sun and is prone to allergies. It will therefore be prudent for these risk factors for keratoconus to be included in the school health education programs.

**Keywords:** keratoconus; risk factor of keratoconus; keratoconus-linked; visual impairment; Cameroon

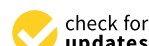



## 1. Introduction

Keratoconus (KC) is a progressive, bilateral, non-inflammatory disorder of the cornea, characterized by central or paracentral, asymmetrical thinning and protrusion of the cornea [1,2]. The word "keratoconus" is derived from two Greek words, *kerato* and *conos*, meaning "cornea" and "cone", respectively, therefore meaning "cone-shaped cornea" [1,2]. KC is a multi-factorial eye disorder associated with hereditary and environmental risks factors such as genetics, ethnicity, ultraviolet (UV) and or sun exposure, eye rubbing, gender, hormones, age, atopy, parental education, floppy eyelids syndrome, poverty, malnutrition and body-weight loss [1,2]. However, recent findings suggest that inflammatory activities could be key in the etiology of keratoconus [1,2]. Undiagnosed KC could affect the sufferer's activities of daily living, productivity and general quality of life (QoL), eventually causing visual impairment if not treated timeously [1,2]. The disease is associated with corneal degenerations and uncorrected refractive errors [1,2].

Globally, KC is one of the leading causes of corneal visual impairment and blindness among students and adolescents, particularly in countries with well-known high prevalence [1,2]. A recent study on KC in Saudi Arabia showed a prevalence of 4.79% among those aged 12–21 years [3]. KC has been reported a low prevalence of 0.03% in Russia, a higher prevalence of 2.3% in central India and 3.18% among Israeli Arabs [4]. Countries reporting higher prevalence rates also have higher exposure to sunlight and UV-rays. Egypt, one of the few African countries reporting on KC in the scholarly literature, reported an incidence of 0.17% in a population-based study [4].

The risk factors of KC could be genetic or environmental. The environmental predictors or factors could function as activators in genetically predisposed persons. Some examples of risk factors for KC are atopy or allergic experience, eye rubbing, ultraviolet (UV) or sunlight exposure, refractive error, positive family history of KC, parental consanguinity, low levels of parental education, age, gender and socio-economic status [4]. Studies have reported contrasting results about gender as a risk factor for KC [5,6]. It has been shown that KC is more prevalent between the second and third decades of life and drastically reduced in prevalence in the fourth, fifth and above decades of life [4].

The Collaborative Longitudinal Evaluation of Keratoconus (CLEK) study showed that almost 50% of the participants rubbed both eyes and 2.2% rubbed one eye strongly [4]. Hashemi et al. [7] found eye rubbing to have a high odd of being associated with KC [3.09 (95% CI: 2.17–4.00)]. The eye rubbing may be triggered by allergies [1.42 (95% CI: 1.06–1.79)] and asthma (1.94 95% CI:1.30–2.58) (2.95 95% CI:1.30–4.59), which were found to be associated with KC. KC has also been shown to be associated with environmental risk factors such as UV-rays and sunlight exposure and socioeconomic factors such as parental education level [7]. Family history of KC and consanguineous marriages have also been reported to be associated with KC, highlighting a genetic link [8].

Cameroon, located in the Central African sub-region, has about 29 million inhabitants [9]. Over 60% of the population is below the age of 25 years with the median age being 18.6 years (males 18.5 years; females 18.7 years). Most inhabitants live in the Western and Northern regions of the country, with the intramural part of the country sparsely populated. There is a variation in the climate of Cameroon, the tropical across the coast to semi-arid and hot climates in the north, depending on the terrain along the length of the country from south to north, respectively. Approximately 37.5% of Cameroonians live below the poverty line [9].

There is currently a paucity of scholarly reports on the risk factors in sub-Saharan African countries, with none reported in Cameroon. As KC is a leading cause of visual impairment globally, it could affect the daily functional vision of those with undetected KC in Cameroon, potentially increasing poverty in the family and reducing their quality of life. This study aimed to determine school population-based risk factors of KC in the West Region of Cameroon. The outcome of this study will help to create awareness of the risk factors linked to keratoconus among the general population and eye health decision makers in Cameroon.

## 2. Method

A quantitative, descriptive, cross-sectional study was conducted in secondary schools in the West Region of Cameroon. Using a stratified cluster random sampling strategy, 3015 secondary school students were recruited as study participants. The modified Keratoconus Risk Investigative Survey (KRIS) questionnaire was used to collect information on the risk profile of KC among the study population.

## 3. Data Analysis

Descriptive analysis was used to provide an overview of general findings, for the risk factors of keratoconus. Estimates were made as the proportion of affected school students and presented with a 95% confidence interval.

Univariable analyses were performed using standard non-parametric and parametric tests (Fisher's exact test if any of the expected frequencies were less than 5) to determine whether age, sex and family history of KC, parental education, consanguinity, atopy or eye rubbing were significantly associated with KC. Multivariate logistic regression analysis was performed to explore the association between KC (the outcome variable) and the independent predictors, which were found significant in the univariable analysis. All the statistical analyses were performed using the statistical package for social sciences (SPSS) version 25. The predictor variables were binary and coded as "1" (KC present) and "0" (KC absent), "1" for males and "0" for females. All tests were two-tailed and $p$-values less than 0.05 were considered statistically significant. Ethical approval was obtained from the Biomedical Research and Ethics Committee (BREC) of UKZN, Durban-South Africa (BREC/00000906/2019), the CAMBIN-ERCC, Yaounde-Cameroon (CBI/456/ERCC/CAMBIN) and gatekeeper letters were obtained from the various levels and study locations in Cameroon. All study participants were treated in accordance with the tenets of the Helsinki Declaration.

## 4. Results

A total of 3015 students, from randomly selected secondary schools in the West Region of Cameroon, participated in the study. Table 1 illustrates the demographic profile of the participants. Most of the students were in the age range between eight and 18 years (93.2%) with a mean age of 13.18 years (SD = 1.08). The majority of the students were female (59.7%). The highest proportion of those who took part in the study (62.1%), were from the West Region, followed by 22.9% from Northwest Region. Most students were Christians (96.8%) and 2.0% were Muslims (Chi-square $p < 0.001$ for all variables).

**Table 1.** Demographic characteristics of the students.

| Characteristics | | Frequency | Percentage (%) | $p$-Value |
|---|---|---|---|---|
| Age (years) | 19–29 | 205 | 6.8 | <0.001 |
| | 8–18 | 2810 | 93.2 | |
| | Total | 3015 | 100.0 | |
| Gender/sex | Female | 1801 | 59.7 | <0.001 |
| | Male | 1214 | 40.3 | |
| | Total | 3015 | 100.0 | |
| Region of origin/Ethnicity | West Region | 1874 | 62.1 | <0.001 |
| | Northwest Region | 686 | 22.9 | |
| | South Region | 192 | 6.4 | |
| | Littoral Region | 96 | 3.2 | |
| | Center Region | 90 | 3.0 | |
| | Remaining five (5) regions | 77 | 2.3 | |
| | Total | 3015 | 100.0 | |
| Religious background | Christianity | 2920 | 96.8 | <0.001 |
| | Muslim | 60 | 2.0 | |
| | Other Religions | 35 | 1.2 | |
| | Total | 3015 | 100.0 | |

Univariate analysis of the various determinant risk factors showed that 2.5% of participants had blood relations with known KC (parents 2.4%, siblings 1.2% and extended relations 96.4%; S.E = 411.9); eye rubbing was reported by 72.2% (very often 36.2%; S.E = 685.906); allergic experience, 40.2% (itching 22.8%; S.E= 502.549); exposure to sunlight (very often 11.4%; >24 h per week 12.3% (S.E = 89.671); parental consanguinity (first cousins = 1.9%, second cousins= 0.8%; S.E = 16,011.139) and parents education up to tertiary level (46.9% and 57.3%), no formal education (1.9% and 1.4%) for mother and father, respectively. The following risk factors were statistically significantly associated with KC: gender, eye rubbing, exposure to sunlight, blood relation having keratoconus, allergic experience. Age and parental consanguinity were not statistically associated with KC (Table 2).

**Table 2.** Significance of risk factors of keratoconus associated with participants.

| Risk Factor | OR (95% CI) | Fisher's Exact Test-Chi-Square Tests; *p*-Value | Statistically Significance (Yes/No) |
|---|---|---|---|
| Gender | 2.024(1.210–19.480) | <0.001 | Yes |
| Age | 1.001(1.000–1.003) | 1.000 | No |
| Eye rubbing | 3.615(3.412–3.830) | <0.001 | Yes |
| Exposed to Sunlight | 2.735(2.609–2.867) | <0.001 | Yes |
| Blood relations having Keratoconus | 41.819(33.287–52.540) | <0.001 | Yes |
| Allergy experienced | 1.070(1.06–1.080) | <0.001 | Yes |
| Parental consanguinity | 0.998(0.997–1.000) | 1.000 | No |

Four parameters were used to estimate the risk factors for KC: eye rubbing, exposure to sunlight, blood relations and allergic experiences. Those who presented with three (3) or more of the above risk factors were considered to be at risk of developing keratoconus. It was observed that 1039 students (34.46%) out of the 3015 study participants questioned were considered to be at risk for developing KC as they had failed 50% or more of the significant risk factors. Regarding the distribution of the risk profile in the general study population, the proportion of gender and age are shown in Table 1 above. Parental consanguinity showed a 1.9% prevalence for first-cousin marriages and 0.8% for second-cousin couples. Concerning parent education levels, for fathers, there was 1.4% for no formal school, 7.9% for primary school, 29.2% for secondary school and 57.3% for tertiary education. For mothers, we had 1.9% for no formal school, 9.9% for primary school, 38.3% for secondary school and 46.9% for tertiary education. The distribution of the other risk factors of KC among the general population is listed in Table 2.

## 5. Discussion

This study aimed to ascertain the risk factors of KC among secondary school students in the West Region of Cameroon. The outcomes were statistically significant and show the importance of the following risk factors of KC: gender, eye rubbing, exposure to sunlight, blood relations with KC, allergies. Eye rubbing had the highest occurrence, thus making it the most important.

Our findings showed the female gender to be a significant risk factor for KC among participants. This result differs from other studies that reported KC to be more prevalent in males than in females [8,10]. Sex hormonal variation could have a link with the micro-biological structural functioning of the cornea, since earlier findings have reported that there are androgen, estrogen, and progesterone receptors on corneal epithelial cells and keratocytes [11]. Moreover, an elevated concentration of estrogen has the devastating effect of thinning the cornea anatomy, thereby associating such a mechanism with the progression of KC [11]. Hence, the level of sex hormones at a given time and stage in one's life, coupled with the interaction with other environmental factors, could lead to an increase in the prevalence of KC in the affected individual or group of gender, be it female or male, over the other. Females, especially those with higher levels of oestrogen, could therefore be more susceptible to developing KC. This risk may increase when coupled with other genetic and environmental risk profiles.

Eye rubbing in this study was the predominant risk factor of KC and had the highest occurrence of 72.2% (95% CI: 70.6–73.8), being very important, among the other risk factors that were statistically significant (Table 2). Out of the 3015 participants, 36% reported rubbing their eyes often, which could be a result of what is referred to as an uncontrolled habit of eye rubbing (UCHER). This is a condition whereby individuals, especially students, cannot help but rub their eyes consciously and or unconsciously, irrespective of the presence or absence of pain perception in so doing. Our research findings show an odd ratio (OR) of 3.6 with a *p*-value less than 0.001 for eye rubbing (Table 2). Several studies reported a strong association between eye rubbing and KC [12,13], whilst other authors reported no

association between eye rubbing and KC [14,15]. The differences among these studies could be related to how UCHER or abnormal eye rubbing was classified. It has been postulated that consistency in eye rubbing could cause corneal epithelial trauma and the release of interleukin-6 and or 8 (IL-6 and or IL-8) and other degenerative biocatalysts that could compromise the corneal integrity, thereby leading to the etiology of KC [16]. Students and the general public should be educated on the symptoms of UCHER or abnormal eye rubbing and advised to visit their eye care practitioner about the possible effects on the etiology of KC. Photographic illustrations on how eye rubbing can lead to keratoconus and visual impairment with advice to consult an eye care practitioner could be used as public health educational tools. This would strengthen their eye health seeking behavior and decision to go for a keratoconus screening. Furthermore, there is a need to conduct similar studies among primary school students in the region, to determine the magnitude and presentation of eye rubbing and its correlation with keratoconus.

For exposure to sunlight and UV-rays, we found a 30.1% (95% CI: 28.5–31.8, OR = 2.7, $p < 0.001$) prevalence. This is similar to other studies conducted in countries with high exposure to sunshine such as Saudi Arabia, India, Israel and Lebanon [14,17–19]. Exposure to a hot and sunny environment has been reported to cause cytotoxicity and thinning of the cornea leading to the pathogenesis of KC due to oxidative stress and damage of highly reactive oxygen derivatives [20]. The study site, Bafoussam, is the capital city of the West Region of Cameroon, has an altitude of about 1500 m above sea level, an average annual temperature of about 21 °C and an average of 8.7 h of exposure to sunlight, which is about 3176 h of sunlight annually [21]. A similar school-based study [14] in Jerusalem showed a high prevalence of KC associated with exposure to sunlight. Jerusalem is about 750 m above sea level, has an annual temperature of 18 °C and has an annual mean of 3397 h of exposure to sunlight. UV from sunlight has devastating effects leading to the etiology of keratoconus, However, it has also been illustrated that controlled and regulated UV-rays in a clinical setting are positively instrumental in the collagen cross-linking procedure for the management of keratoconus and other corneal-related pathologies [22]. Keratoconus could lead to visual impairment and reduced productivity of an affected individual, with a long-term negative impact on the quality of life of the individual and his/her contribution to society. There is therefore a need to educate students and the general population on the various ways to protect themselves from the uncontrolled UV-rays that have harmful effects and could lead to keratoconus. They should reduce the number of hours being exposed to UV-rays and sunlight mostly between 10 am and 4 pm in the day, wear UV-protective eyeglasses when walking under the sun or when using an electronic gadget such as laptops without UV-rays screen protection, wear brimmed hats and eat enough servings of vitamin A and C to reduce the harmful effects of UV-rays.

Blood relation or positive family history of KC was statistically significantly associated with a prevalence of the disease. However, it was 11 times and 5 times lower than eye rubbing and exposure to sunlight, respectively (Table 2). Other studies reported similar results [14,23]. In our view, the reason for the statistical significance of this risk factor could be the activation of genetic factors that were inherited, especially by first-degree relatives. The genetic factors could be triggered by some environmental factors such as eye rubbing and exposure to sunlight, especially if the blood relatives are living in the same geographical location. In addition to genetic testing and counselling on the need for eye examinations of family members when a diagnosis of KC is made, public education should be undertaken to highlight blood relation as a risk factor of KC in the Cameroon setting.

Allergic experiences, in this study, were the third most significant and important risk factor of KC (Table 2). Nemet et al. [24] also found an association between allergic experiences and KC. In this study, allergic experiences as risk factors of KC were slightly more predominant than exposure to sunlight and about two times lower than eye rubbing. Among those reporting allergic experiences, we found a high proportion (22.8%) also reporting itching. If the itching occurs in the eye, then it could lead to eye rubbing and

subsequently develop into KC. Further research is required to establish the occurrence of the various allergic experiences and their possible effect on the etiology of keratoconus.

The parental educational level as a risk factor of KC was insignificant, which could be due to the fact that most parents have access to secondary and tertiary education in the country. Age and parental consanguinity were also not significant risk factors in this study. For age, this could be a result of a wider variation in the study participants and the onset of KC may occur in many younger children than in the usual age of onset of KC [4]. The reason that consanguinity was not significant could be due to the majority of the local tribes and cultures prohibiting intermarriage between relatives of any degree. However, previous studies [8] have found that consanguinity does pose a risk for keratoconus.

The result of the study has revealed that 34.46% of students were at risk to develop KC, which is similar to a study by Almusawi et al. 2021 [25]. The profile of a child at risk of developing KC in Cameroon is one that engages in eye rubbing, has a family member with KC, spends more than eight hours per week in the sun and is prone to allergies. It will therefore be prudent for these risk factors for keratoconus to be included in the school health education programs.

## 6. Limitations

The limitation of this study was that it was based on self-reports and is therefore subjected to all the limitations of self-reported inquiries such as recall bias, etc.

## 7. Conclusions

Eye rubbing was the most significant risk factor, followed by allergic experiences and exposure to sunlight. These findings support the evidence that the etiology of KC is multifactorial with eye rubbing being the most significant factor in this cohort. There is a need to address eye rubbing among students to minimize the risk of KC. Finally, with 34.46% of study participants being at risk of developing keratoconus, the researchers recommend attention to KC management be provided by public health authorities in Cameroon.

**Author Contributions:** Conceptualization: E.N.A., V.R.M. and K.P.M.; data Curation: E.N.A.; formal analysis: E.N.A., V.R.M. and K.P.M.; methodology: E.N.A., V.R.M. and K.P.M.; supervision: V.R.M. and K.P.M.; writing—original draft: E.N.A.; writing—review and editing: V.R.M. and K.P.M. All authors have read and agreed to the published version of the manuscript.

**Funding:** The authors declare that no form of funding was received for this study.

**Institutional Review Board Statement:** The study was conducted according to the guidelines of the Declaration of Helsinki and approved by the Research and Ethics Committee of the University of KwaZulu-Natal (UKZN), (date of approval: 14th August 2020; Reference number: BREC/00000906/2019), the CAMBIN-ERCC, Yaounde-Cameroon (date of approval: 21st July 2020; Reference number: CBI/456/ERCC/CAMBIN) and gatekeeper letters were obtained from the various levels and study locations in Cameroon. Permission to access patients was obtained from the Ministry of Education in Cameroon.

**Informed Consent Statement:** Written informed consent was sought from the parents/guardians of participating school children; the children also gave their assent. Participants were informed that they are free to withdraw at any stage of the study if they will no longer wish to continue in the study. The identity of all participants is anonymized.

**Data Availability Statement:** The datasets used and/or analyzed during the current study are available from the corresponding author upon reasonable request.

**Acknowledgments:** The authors wish to thank Enow Samuel, Tataw, George Moyo and Kepang Nanseu Evrad Melvin, for assistance with statistical analyses.

**Conflicts of Interest:** The authors have no financial disclosures to make and no conflict of interest to declare.

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
