# Peer review of "Risk Profile of Keratoconus among Secondary School Students in the West Region of Cameroon"

_2411-5150_

Round 1
Reviewer 1 Report
You should describe how keratoconus is diagnosed.
Author Response
Dear Sir, Madam,
Thank you for your kind words and time.
With respect to this manuscript, a standardized questionnaire was administered, to determine the risk factors associated to keratoconus.
There is a different manuscript that focuses on classification, prevalence and epidemiological profile of keratoconus. Hence we had a brief description on KC in the introduction of this manuscript, in order not to another section we had mapped out for another manuscript.
I hope we have answered your questions.
Thanks once again for your review and comments, we deeply appreciate.
Best Regards
Reviewer 2 Report
This study explores risk factors of keratoconus amongst 3015 secondary school students in the West Region of Cameroon using questionnaires. Gender, eye rubbing, exposure to sunlight, blood relation having keratoconus, allergic experience, refractive error and parent education level were all found to be statistically significant risk factors. I have a few comments and suggestions for improvement as follows:
1) Page 3, Lines 217-218. What is mentioned in the text here does not seem to match the numbers in the table. The authors reported that 93.2% were between the ages of 8-28 years old, however Table 1 indicates that 93.2% were between 8 and 18. If there is an error here, it should also be adjusted in lines 22-23 in the abstract.
2) How was information on refractive error obtained? Was this self-reported by the participants? If self-reported, I feel that refractive error would be difficult to obtain accurately from participants as presumably many would not be aware of how their refractive error is correctly classified.
3) Can the authors provide some extra details in terms of what was defined as allergic experiences? The authors mentioned itchy eyes as one component, but were there any other components that were included as an allergic experience?
4) In this study the authors identified a number of risk factors through surveys and questionnaires. Are there any plans to supplement these findings with more quantitative measures such as corneal topography?
5) In the demographic information, can the authors provide details on how many of the sample have keratoconus and how many don’t?
Author Response
Dear Sir, Madame,
Thank you very much for your kind review and comments.
1)It is a typographical error. The proportion of 93.2% were between eight to 18 as illustrated on Table 1, and not eight to 28 years, it has been corrected, thanks very much.
2) Thank you very much for your question. This manuscript is one of a series, from the project work that we did. The Refractive error just like the presenting visual acuity was obtained by trained data collectors using VA charts and autorefractokeratometer respectively.
3)Thanks again for raising this issue. Allergic experience as we are aware involves itching, asthma etc. Itching had the highest proportion. The allergic experiences was mention in the introduction and our interest was on the one with the highest proportion, which is itching and how it has relevance to our findings.
4) Thank you very much for this question, as earlier mentioned, this study is part of a series of our research project. Yes there are subsequent manuscripts on the clinical phase of the research project, thanks very much.
5) Thanks again for this question, as earlier mention in point two, this manuscript is part of a series of a research project. Furthermore, we used a standardized form for data collection to determine the risk factors of keratoconus among study participants. There is another manuscript that focuses on the classification, prevalence and epidemiological profile of KC.
Thanks very much for all the issue raised, we deeply appreciate and we hope we have answered your questions.
Best Regards